# Temporal dynamics of SARS-CoV-2 shedding in feces and saliva: a longitudinal study in Norfolk, United Kingdom during the 2021–2022 COVID-19 waves

Lee Kellingray,[1] George M. Savva,[1] Enriqueta Garcia-Gutierrez,[1,2] Jemma Snell,[1] Stefano Romano,[1] Daniel Alejandro Yara,[1] Annalisa Altera,[1] Leonardo de Oliveira Martins,[1] Chloe Hutchins,[1] David Baker,[1] Antonietta Hayhoe,[1] Christian Hacon,[3] Ngozi Elumogo,[4] Arjan Narbad,[1] Lizbeth Sayavedra[1]

**ABSTRACT** Severe acute respiratory syndrome coronavirus 2 (SARS-CoV-2) was originally described as a respiratory illness; however, it is now known that the infection can spread to the gastrointestinal tract, leading to shedding in feces potentially being a source of infection through wastewater. We aimed to assess the prevalence and persistence of SARS-CoV-2 in fecal and saliva samples for up to 7 weeks post-detection in a cohort of 98 participants from Norfolk, United Kingdom using RT-qPCR. Secondary goals included sequencing the viral isolates present in fecal samples and comparing the genetic sequence with isolates in the saliva of the same participant. Furthermore, we sought to identify factors associated with the presence of detectable virus in feces or saliva after a positive SARS-CoV-2 test. Saliva remained SARS-CoV-2-positive for longer periods compared to fecal samples, with all positive fecal samples occurring within 4 weeks of the initial positive test. Detectable virus in fecal samples was positively associated with the number of symptoms experienced by the individuals. Based on the genome sequencing and taxonomic classification of the virus, one donor had a distinct strain in feces compared to saliva on the same collection date, which suggests that different isolates could dominate different tissues. Our results underscore the importance of considering multiple biological samples, such as feces, in the detection and characterization of SARS-CoV-2, particularly in clinical procedures involving patient fecal material transplant. Such insights could contribute to enhancing the safety protocols surrounding the handling of patient samples and aid in devising effective strategies for mitigating the spread of coronavirus disease.

**IMPORTANCE** This study provides critical insights into the dynamics of severe acute respiratory syndrome coronavirus 2 (SARS-CoV-2) shedding in fecal and saliva samples, demonstrating that while viral RNA is detectable shortly after diagnosis, its prevalence declines rapidly over the course of infection. Detection was more common among individuals with more concurrent symptoms, emphasizing the potential influence of symptom burden on viral persistence. By analyzing a United Kingdom-based cohort, this study fills a significant gap in the literature, which has largely focused on Asian and North American populations, offering a geographically unique perspective on viral shedding dynamics. Our findings contribute to a globally relevant understanding of SARS-CoV-2 shedding by revealing differences in shedding durations compared to studies from other regions. These differences highlight the need for geographically diverse research to account for variations in genetic background, immune response, and healthcare practices.

**KEYWORDS** COVID-19, SARS-CoV-2

**Peer Reviewer** Pablo Tsukayama, Universidad Peruana Cayetano Heredia, Lima, Peru

Address correspondence to Lizbeth Sayavedra, lizbeth.sayavedra@quadram.ac.uk.

The authors declare no conflict of interest.

See the funding table on p. 12.

While primarily known for causing respiratory disease, there is now growing evidence suggesting that severe acute respiratory syndrome coronavirus 2 (SARS-CoV-2) can impact the gastrointestinal system (1–3). This has implications not only for the acute phase of the infection but also for long-term health outcomes.

SARS-CoV-2 enters human cells by binding to the angiotensin-converting enzyme 2 (ACE2) receptor, which is not only prevalent in the lungs but also in the ileum and colon (4, 5). This distribution of ACE2 receptors potentially explains the ability of SARS-CoV-2 to colonize the gastrointestinal tract, which could lead to gastrointestinal disease. Indeed, post-coronavirus disease (COVID-19) syndrome has been associated with ongoing gastrointestinal symptoms in approximately 10.8% of patients, with an increased risk of developing functional gastrointestinal illnesses, such as irritable bowel syndrome and inflammatory bowel disease (6).

The detection of SARS-CoV-2 in fecal samples of infected individuals has raised concerns about feces as a potential transmission route. This concern is further heightened by historical data from viruses like SARS and Middle East respiratory syndrome, which were known to be transmitted through fecal material and other bodily fluids (7). SARS-CoV-2 has demonstrated a tendency to persist in the gut, even after respiratory symptoms subside and nasopharyngeal swabs test negative (8). Studies have reported varying durations of viral shedding in feces, ranging from 7 (9) to over 70 days in children, using RT-qPCR (10). In addition, studies have documented SARS-CoV-2 fecal shedding up to 5 weeks after respiratory samples tested negative (11), with the longest reported duration being 48 days after negative upper respiratory samples (12). This prolonged fecal shedding has significant implications for transmission dynamics, environmental contamination, and public health strategies. Consequently, fecal testing has emerged as a valuable epidemiological surveillance tool worldwide (13–16).

To further our understanding of SARS-CoV-2 persistence in different bodily fluids, we conducted a longitudinal study in Norfolk, United Kingdom (UK) during the 2021–2022 COVID-19 waves. Our primary aim was to estimate the prevalence and persistence of SARS-CoV-2 in fecal and saliva samples for up to 7 weeks following a positive SARS-CoV-2 test result. Additionally, we aimed to identify the viral strains present in feces and compare these with strains detected in saliva. Finally, we sought to identify factors that predicted the presence of detectable virus in feces following a positive SARS-CoV-2 test.

## MATERIALS AND METHODS

### Participants

Recruitment was conducted from both community and hospital settings through the Quadram Institute website, posters at COVID-19-associated sites, a social media campaign, local news outlets, the Quadram Institute Clinical Research Facility, and a National Health Service (NHS) research nurse. Hospital patients were identified through clinical records and referrals. Potential participants were included if they met the following criteria: (i) aged 18 years or older, (ii) lived or worked within a 40 mi radius of Norwich, the Norfolk and Norwich University Hospital or James Paget University Hospital (JPUH), and (iii) had been confirmed as SARS-CoV-2-positive by the NHS testing service or the COVID-19 Norwich Testing Initiative. Exclusion criteria included individuals admitted to an NHS Intensive Care Unit, except for non-ventilated High-Dependency Unit patients who could give informed consent. Additionally, individuals unable to provide written informed consent and those without a positive SARS-CoV-2 test were excluded.

On recruitment, participants provided information on their date of birth, sex, and ethnicity. At each weekly sample collection, they reported their COVID-19 symptoms based on a structured questionnaire, which included the following symptoms: fever, persistent dry cough, sore throat, tiredness, shortness of breath, muscle or joint ache, diarrhea, vomiting, and loss of taste or smell. Participants also reported their current vaccination status (whether they had received a full or partial course of a COVID-19 vaccination and which vaccine(s) they had received).

A set of feces and saliva samples was collected at recruitment, followed by an additional three sets collected at weekly intervals thereafter. Participants were provided with identical fecal and saliva collection kits and instructions for each of the three weekly follow-up collections, and they were asked to collect the saliva and the fecal samples in the same way as the first collection. Each sample set also had a shorter questionnaire reduced in size from the initial one to collect updates on vaccination status and ongoing symptoms.

## Sample size

We aimed to recruit between 100 and 200 participants. The target minimum sample size of 100 participants was determined by setting the required precision of the estimate of baseline prevalence of COVID-19 detectable in feces. We anticipated a baseline positive rate between 10 and 90%, and in that range, we would be able to estimate the prevalence at baseline to within at least ±10 percentage points (95% confidence interval, CI). Further considerations, if sufficient positive samples were collected, were the precision of estimates for the rate of decline in prevalence over time since diagnosis and the power to identify predictors of feces positivity, but these were based on a higher baseline prevalence than was observed for this cohort, as previous studies determined that approximately 60% of COVID-19-positive subjects had detectable loads of SARS-CoV-2 in associated fecal samples [23].

## Viral inactivation of feces and saliva samples

SARS-CoV-2 inactivation was performed at Containment Level 3 (CL3) following the Advisory Committee on Dangerous Pathogens classification of SARS-CoV-2 as a hazard group 3 pathogen. To inactivate SARS-CoV-2 from feces, 100 mg of feces per donor was transferred into MagMAX bead tubes containing 800 µL of Thermo Fisher MagMAX lysis buffer. The screw caps were tightly fastened, and the bead tubes were subjected to heat treatment for 15 min at 68°C in a heat block, serving as an additional SARS-CoV-2 inactivation step beyond the manufacturer's protocol. The bead tubes were vortexed for 10 s to homogenize the feces samples with the lysis buffer. Following homogenization, heat-treated bead tube samples were vortexed at a minimum speed of 2,500 rpm for 10 min to ensure sample lysis. Samples were centrifuged for 2 min at $14,000 \times g$. After centrifugation, $2 \times 400$ µL aliquots of lysate were dispensed into microcentrifuge tubes. The inactivated sample lysate was stored at −20°C until further processing. Nucleic acid extraction was completed using the MagMAX Microbiome Ultra Nucleic Acid Isolation Kit (A42357; Thermo Fisher, UK) according to the manufacturer's instructions.

For saliva samples, 120 µL of saliva was transferred to a 2 mL microcentrifuge tube. To each tube, 330 µL of Promega Maxwell lysis buffer master mix was added. The sample and buffer were vortexed for approximately 10 s to ensure thorough homogenization. Microcentrifuge tubes were then securely sealed and placed in a heat block at 56°C for 10 min. Following incubation, the tubes were removed from the heat block and allowed to cool to room temperature. Inactivated sample lysates were removed from the CL3 laboratory to a CL2 laboratory for further processing. Nucleic acids from saliva samples were extracted using the Maxwell RSC Viral Total Nucleic Acid Purification Kit (AS1330, Promega, UK) according to the manufacturer's instructions.

## Detection of SARS-CoV-2 by qPCR reaction

Detection of SARS-CoV-2 in feces was conducted as previously described (17). Briefly, 5 µL of the extracted nucleic acids was added to a 15 µL Master Mix containing 1.5 µL of primer sets N1 and N2 (Integrated DNA Technologies, Belgium, 10,006,713) at 6.7 µM, 10 µL of 2× Probe 1-Step Go No Rox (PCR Biosystems) (18), 1 µL of 20× RTase Go, and 2.5 µL of nuclease-free water (Sigma-Aldrich, UK). The reactions were conducted in triplicate on a StepOnePlus Real-time PCR System (Applied Biosystems) under the following conditions: 50°C for 10 min, 95°C for 2 min, followed by 45 cycles of 95°C for 5 s,

55°C for 30 s, and concluding with 40°C for 30 s. Samples with positive amplification in at least one RT-qPCR technical replicate were considered positive.

## SARS-CoV-2 whole-genome sequencing

Saliva and feces samples that tested positive for SARS-CoV-2 were further processed to enable whole-genome sequencing of the virus. cDNA was used for a multiplex PCR, which specifically targeted the SARS-CoV-2 genome. The PCR amplicon concentrations were normalized and used for genome sequencing as described previously (19). Amplicons were sequenced on a NextSeq 2000 instrument using 10mer UDI barcoded primers.

## Bioinformatics analyses

A total of 23 out of 43 saliva and three out of 17 feces-derived sequences passed basic quality control as defined by the COVID-19 Genomics UK Consortium (COGUK) (20) and were aligned with pangolin v4.3 (21) against their early, anonymized SARS-CoV-2 lineage A reference genome. These 26 sequences were compared to a curated alignment of all sequences submitted to COGUK (22) as of June 2023. The 2,999,160 COGUK sequences were labeled "global" in contrast to our 26 "local" sequences, even though all were collected and sequenced in the UK. We extracted the eight closest global neighbors of each of our sequences, prioritizing exact ACGT matches and the potentially distinct eight neighbors with the least number of partial mismatches using uvaia (23). The final data set comprised 288 global sequences, which, together with the local sequences, had their lineages classified with pangolin v4.3. We inferred their maximum likelihood phylogeny with IQTREE2 v2.2.2.7 (24) under the HKY substitution model (25) with gamma rate heterogeneity (26) over all 313 sequences.

## Statistical analyses

We described the proportion of individuals with detectable SARS-CoV-2 in feces and saliva samples over time following the first positive COVID-19 test. We then used regression models to identify predictors of SARS-CoV-2 detection in either sample type.

For descriptive analyses, the index date defined as the date of the positive COVID-19 test was categorized into weeks. During each week, the prevalence of SARS-CoV-2 in feces and saliva samples was estimated.

A multilevel logistic regression model was then used to investigate the effects of various exogenous factors on the detection of SARS-CoV-2 in each sample type using as fixed effects the vaccination status, calendar date, participant age, days since the positive test, cohort, and sample type. Nested random effects were represented by individual donors and measurement occasion within donor. Several candidates for the form of the relationship between the log odds of sample positivity and days since positive test were considered, including linear, logarithmic, cubic spline, and quadratic relationships, with the best model selected by Akaike Information Criterion and validated by comparing marginal means with the observed proportion over time. Marginal means over time since the first COVID-19 tests were calculated from these models to estimate the rate of feces and saliva sample positivity over time in an average participant.

A second model incorporating symptom counts was also developed to explore the link between symptom persistence, the number of symptoms, and detection of virus in feces and saliva.

Independent logistic regression models adjusting for time since diagnosis were used to assess any residual correlation between SARS-CoV-2 detection in fecal and saliva samples, as well as between serial fecal and saliva samples. Models were validated by visually inspecting the predictions versus the observed counts each week since the COVID-19 test for both sample types.

Multiple imputation was used where data were incomplete. In particular, the date of the index COVID diagnosis was not known for 29 participants. Predictive mean matching

was used to impute all missing variables, with the unit of analysis for imputation models being the individual participant to ensure the structure of the data set was preserved. Twenty imputations were used.

All statistical analyses were conducted using R version 4.3.2. Regression models were estimated using glmmtmb (v.1.1.9), while predictions were calculated using emmeans (v1.10.3). Multiple imputation was implemented using mice (v.3.16.0). Pooling of imputation models was conducted by mice via emmeans for predictions or gtsummary (v 1.7.2) for regression coefficients. The analysis data set and code are openly available at https://github.com/quadram-institute-bioscience/cops.

## RESULTS

### Participant recruitment

A total of 120 participants were initially recruited for the study, with 100 providing at least one sample, and 98 included in the final analysis (Table 1). The median time between the first positive COVID-19 test result and the first sample collection was 13 days. A total of 79 (81%) participants returned all requested samples collected between 3 and 56 days after their index positive test.

Participant exclusion was attributed to factors, such as non-return of samples or questionnaires, as well as withdrawal from the study due to reasons, including concerns about mental health, being too unwell to continue, or relocation from the study area.

### Prevalence of SARS-CoV-2 in saliva and fecal samples

For fecal samples, 20 out of 357 samples tested were positive from 15 different participants. For saliva, 48 out of 356 samples were positive from 36 participants (Table 2). Only four sample sets were acquired within the first 7 days post-positive test, but from these, three were positive in feces and saliva. Subsequently, the rate of detection in both sample types declined rapidly, and of the 56 fecal samples collected more than 3 weeks after the index positive test, none were positive for SARS-CoV-2.

### Predictors of fecal and saliva sample positivity over time

Figure 1B and Table 3 show the temporal changes in the prevalence of sample positivity. The model that best described these data showed a linear relationship between the log odds of sample positivity and the log of days since the initial positive test, suggesting a power–curve relationship between the probability of sample positivity and days since

**TABLE 1** Characteristics of participants included in the study[a]

| Characteristic | Community, $N = 81$[b] | JPUH, $N = 17$[b] |
|---|---|---|
| Total number of samples included per participant | | |
| 2 | 4 (4.9%) | 2 (12%) |
| 4 | 3 (3.7%) | 2 (12%) |
| 6 | 4 (4.9%) | 3 (18%) |
| 7 | 1 (1.2%) | 0 (0%) |
| 8 | 69 (85%) | 10 (59%) |
| Age (years median, IQR) | 48 (40, 56) | 51 (49, 67) |
| Missing | 1 | 1 |
| Ever vaccinated | 69 (91%) | 7 (100%) |
| Missing | 5 | 10 |
| Days between diagnosis and first sample (median, IQR) | 13.0 (10.0, 18.3) | 14.0 (9.0, 15.0) |
| Missing | 21 | 8 |
| Date of COVID diagnosis (range) | 23 December 2021 to 5 February 2022 | 3 November 2021 to 14 March 2022 |
| Missing | 21 | 8 |
| Symptom score at baseline (median, IQR) | 3 (4–6) | 5 (5, 6) |

[a]The percentages in this table were calculated based on the number of individuals who returned their questionnaires.
[b]n (%).

**TABLE 2** Number and proportion of positive samples stratified by the number of completed weeks since the index positive test[a]

| Weeks since positive test | Number of sample sets | Positive fecal samples | Positive saliva samples |
|---|---|---|---|
| 0 | 4 | 3 (75%) | 3 (75%) |
| 1 | 38 | 5 (13%) | 12 (32%) |
| 2 | 54 | 4 (7.4%) | 5 (9.3%) |
| 3 | 60 | 2 (3.3%) | 5 (8.3%) |
| 4 | 56[b] | 0 (0%) | 1 (1.8%) |
| 5 | 29 | 0 (0%) | 2 (6.9%) |
| 6 | 10 | 0 (0%) | 1 (10%) |
| 7 | 1 | 0 (0%) | 0 (0%) |
| 8 | 1 | 0 (0%) | 0 (0%) |
| Unknown | 104 | 6 (5.8%) | 19 (18.3%) |

[a]n (%).
[b]One saliva sample missing.

infection, although similar fits and results were obtained with models using quadratic functions and cubic splines to represent this relationship.

Saliva samples were approximately four times more likely to test positive for the virus compared to fecal samples [odds ratio (OR) = 3.85, 95% CI = 1.9 to 7.8; $P < 0.001$]. While there was no significant interaction between sample type and time since the initial positive test ($P > 0.9$), the number of positive fecal samples was small, and this would be difficult to detect.

Vaccination status also significantly influenced virus detection rates, with vaccinated individuals showing a lower likelihood of testing positive (OR = 0.22, 95% CI = 0.05 to 0.90; $P = 0.035$) compared to those who were unvaccinated (Table 3), although the number of non-vaccinated participants in the sample was small. There was no significant association between virus detection probability and factors, such as participant age, calendar date, or patient source (community vs. hospital).

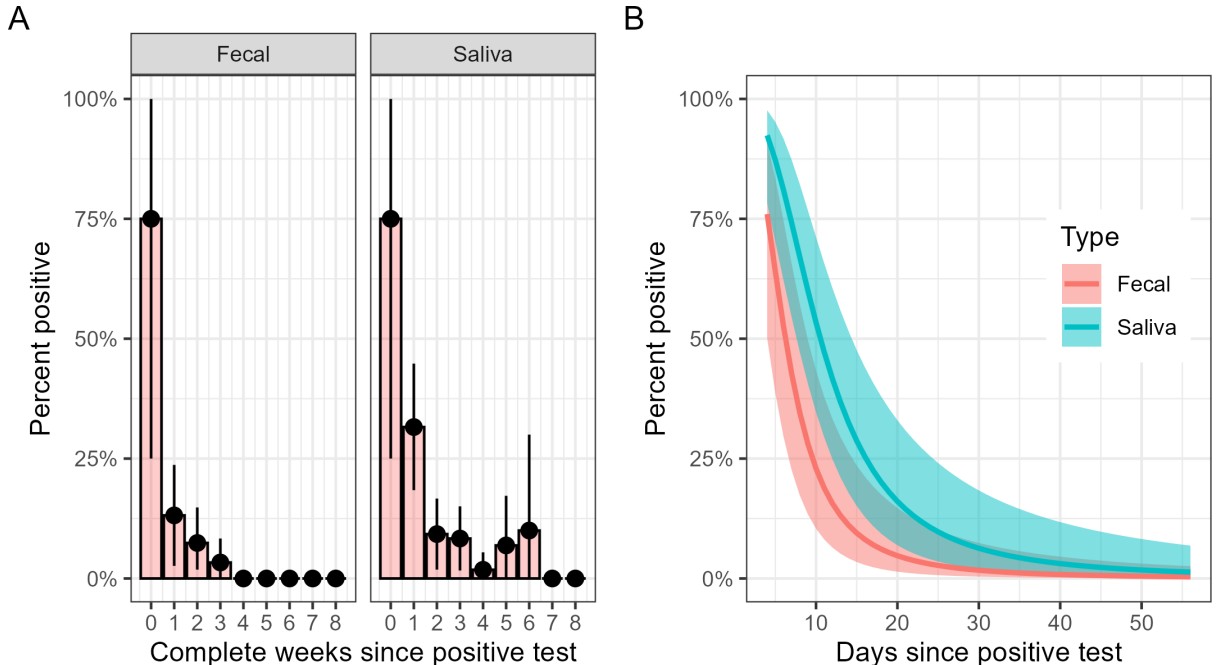

**FIG 1** Prevalence of SARS-CoV-2 in fecal and saliva samples over time since index positive test. (A) The proportion of observed positive samples stratified by completed weeks since the index positive test using only data where the time since diagnosis is recorded. (B) Modeled estimates, with ribbons corresponding to 95% confidence intervals with multiple imputations used to incorporate cases with missing data.

**TABLE 3** Multivariable mixed-effects logistic regression model showing the effects of patient characteristics and time since diagnosis on the odds ratio (OR) of positive samples at follow-up[c]

| Characteristic | Participant characteristics only | | | Model including concurrent symptom count | | |
|---|---|---|---|---|---|---|
| | OR | 95% CI[a] | *P*-value | OR | 95% CI[a] | *P*-value |
| Vaccinated | 0.22 | 0.05, 0.90 | 0.035[b] | 0.21 | 0.06, 0.74 | 0.016[b] |
| Date of diagnosis (per quarter year) | 0.40 | 0.08, 1.97 | 0.256 | 0.43 | 0.09, 2.16 | 0.303 |
| Age (per year) | 0.99 | 0.96, 1.02 | 0.469 | 0.98 | 0.96, 1.01 | 0.259 |
| Log (days since diagnosis) | 0.08 | 0.03, 0.19 | <0.001[b] | 0.13 | 0.05, 0.29 | <0.001[b] |
| Hospital vs. community | 1.82 | 0.67, 4.96 | 0.237 | 1.79 | 0.72, 4.41 | 0.206 |
| Saliva vs fecal sample | 3.85 | 1.89, 7.84 | <0.001[b] | 3.72 | 1.92, 7.18 | <0.001[b] |
| Concurrent symptoms (per symptom) | | | | 1.23 | 1.02, 1.47 | 0.026[b] |

[a]CI = confidence Interval, OR = odds ratio.
[b]Considered significant.
[c]Concurrent symptom count refers to the sum of different symptoms reported, which included the following: fever, persistent dry cough, sore throat, tiredness, shortness of breath, muscle or joint ache, diarrhoea, vomiting, and loss of taste or smell.

## Relationship between the detection of the virus in samples with symptom counts

Figure 2 shows the relationship between the number of symptoms reported and completed weeks since the index positive test.

Symptom count was significantly associated with virus detection in the multivariate model, with an odds ratio of 1.23 per symptom point (95% CI 1.02 to 1.47; *P* = 0.026), indicating that individuals with more symptoms were more likely to have detectable virus in fecal or saliva samples after adjusting for days since the initial positive test and other predictors.

## Correlation between fecal and saliva samples over time

In some instances, a negative sample was followed by a positive sample. For example, of the eight participants with a positive saliva sample at the second time point, only five had a positive saliva sample at the first time point. Similarly, among the four participants with a positive fecal sample at the second time point, only three had positive fecal samples at the first time point. This may be due to re-infection during the sampling period, or it could indicate that the level of viral particles present in the samples was lower than the limit of detection of our assay.

There was some evidence for a correlation between the presence of COVID-19 in saliva and fecal samples (OR = 1.9, *P* = 0.502) after adjusting for time since the index test. However, there was little evidence for an association between positive fecal samples at the first and second follow-up points (OR = 0.855, *P* = 0.938) likely due to the small number of positive fecal samples at the second follow-up. The relationship between positive saliva samples at the first and second time points was higher (OR = 7.9; *P* = 0.073), albeit not statistically significant at *P* < 0.05.

## Phylogenetic classification of the detected SARS-CoV-2

From the 20 SARS-CoV-2 positive fecal samples and 48 saliva samples we collected, we successfully sequenced and classified three fecal and 27 saliva samples (Fig. 3; Fig. S1). The Omicron (BA.1-like and unassigned) variant was the most frequently detected in saliva samples (*N* = 12 donors), followed by the Delta (B.1.617.2-like) in five donors, Delta (AY.4-like) in four donors, and Alpha (B.1.1.7-like) in three donors. The three fecal samples for which we could sequence the SARS-CoV-2 variant belonged to the Delta type. These findings highlight the diversity of SARS-CoV-2 variants circulating in the Southeast of England during the study period. In only one case did the variants in the saliva and fecal sample differ, with the Alpha variant (B.1.1.7-like) in saliva and the Delta variant (B.1.617.2-like) in feces.

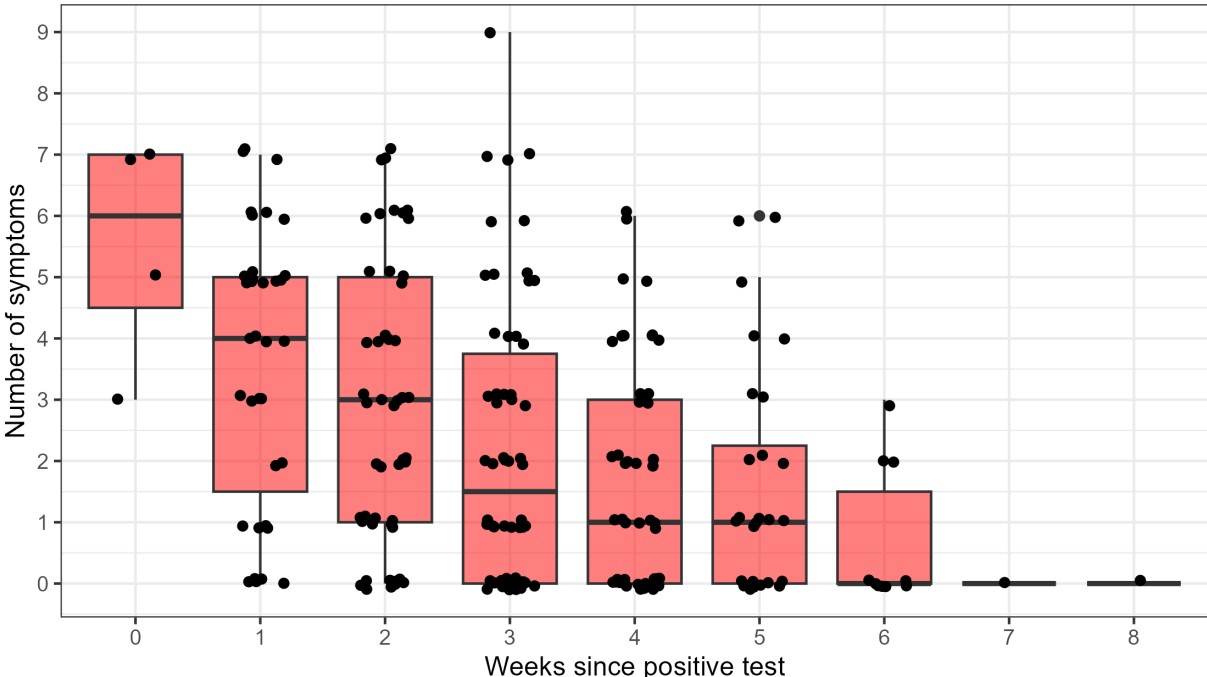

**FIG 2** Distribution of the number of reported COVID-19 symptoms over the number of completed weeks since positive test (boxes represent median and interquartile range).

## DISCUSSION

We have estimated the prevalence of SARS-CoV-2 in saliva and fecal specimens in the weeks following a positive COVID-19 test and explored factors associated with prevalence and persistence. We found that while a high proportion of participants had detectable SARS-CoV-2 in fecal and saliva samples in the week following a positive test, this prevalence decreased rapidly over time (Fig. 1). SARS-CoV-2 was not detected in any of our collected fecal samples by the end of the fourth week. In contrast, saliva samples showed a slightly extended persistence, with detectable virus in some cases up to 6 weeks post-diagnosis. This pattern underscores the potentially shorter duration of SARS-CoV-2 shedding in feces compared to saliva, which has implications for understanding transmission dynamics and monitoring viral shedding in different biological matrices.

### The prevalence and persistence of SARS-CoV-2 in feces vary across populations and time

The presence of SARS-CoV-2 in the feces of this cohort aligns with previous studies demonstrating SARS-CoV-2 shedding in feces (9, 27). However, the persistence of viral shedding in feces appears to vary greatly among patients and cohorts (28). The rapid decline in fecal sample positivity observed in this study is consistent with a meta-analysis of 35 studies ($N = 1,636$ participants), which reported an average fecal shedding duration of approximately 21.8 days compared to 14.7 days in respiratory samples—a shorter persistence than reported in studies with prolonged viral detection (27). A similar observation was made by van Doorn et al. who performed a qualitative analysis of 95 studies ($N = 2,149$ participants) and observed that those patients whose fecal samples tested positive for SARS-CoV-2 ($N = 282$) remained positive for a mean of 12.5 days and up to a maximum of 33 days after respiratory samples tested negative (29). While Scaglione et al. (30) documented fecal shedding up to 126 days and Natarajan et al. (31) for up to 7 months, this persistence may reflect differences in sampling intervals, testing sensitivity, or even individual immune responses. Our findings indicate a shorter viral shedding duration in feces at least in the majority of cases, which may suggest a lower

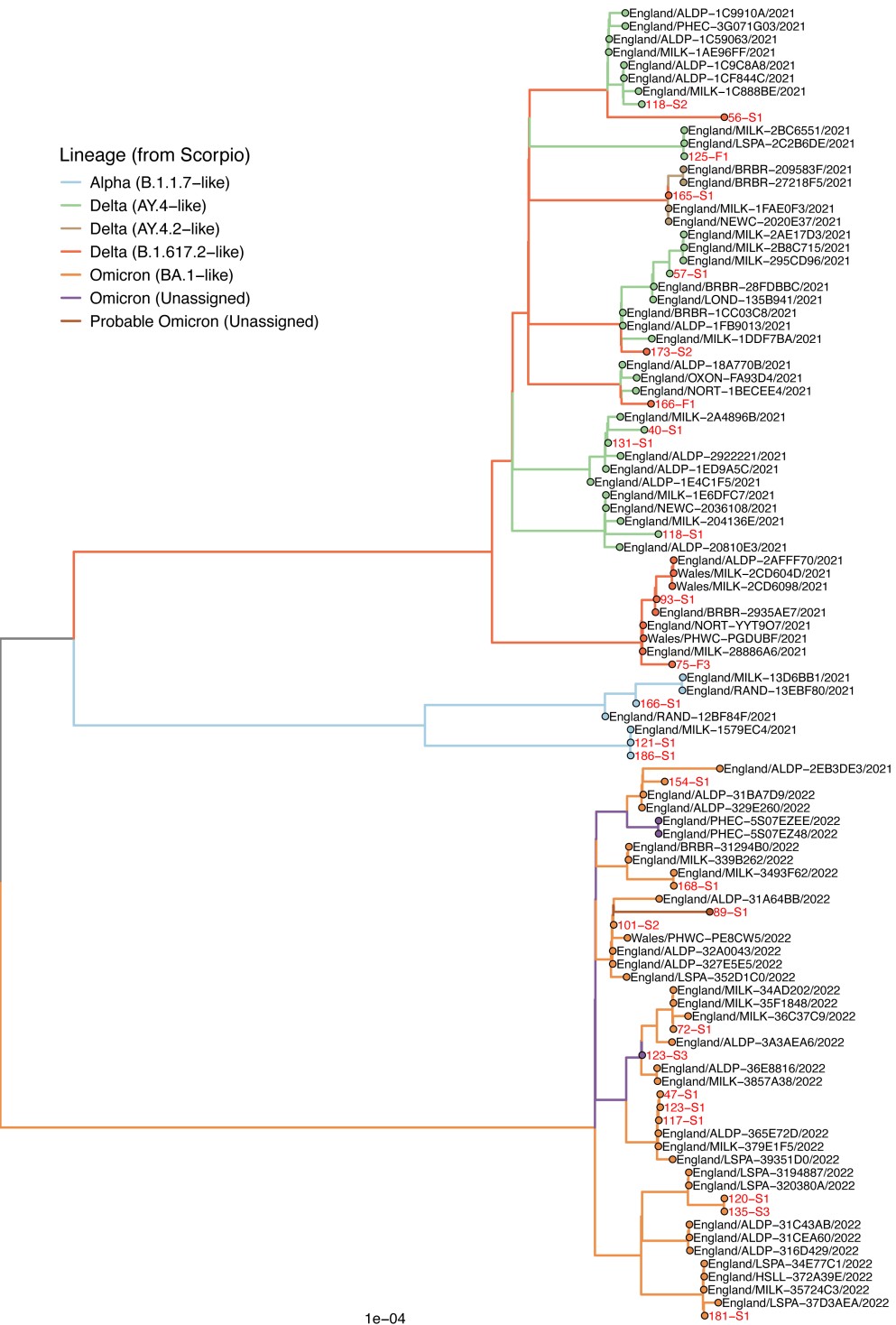

**FIG 3** Phylogenetic tree of SARS-CoV-2 sequences recovered from feces and saliva.

risk of prolonged fecal transmission within this study population. This difference could be caused by several factors, including the widespread rollout of vaccines, differences in sampling methods, or the prevalence of specific SARS-CoV-2 variants with potentially altered shedding dynamics.

In contrast to studies reporting SARS-CoV-2 persistence in feces across diverse cohorts, including asymptomatic and mildly symptomatic individuals, as well as children (32), our results suggest that SARS-CoV-2 may clear more quickly in typical adult

cases with predominantly mild to moderate symptoms. Recent genomic epidemiology research in Norfolk, UK (33) identified extensive SARS-CoV-2 lineage diversity, with shifts in dominant variants occurring on a slightly different timeline than national and global trends. Since viral genetics influence shedding dynamics, variations in lineage predominance may have contributed to the shedding durations observed in our study. Factors, such as age, disease severity, and immune response, as well as population-specific dynamics, such as vaccination coverage, healthcare interventions, and demographic characteristics (e.g., Norfolk's older population compared to the rest of the UK), including re-infection, could account for the variability in shedding duration reported across studies. For instance, a Chinese cohort study noted that SARS-CoV-2 persisted in feces for about 7 days after respiratory samples tested negative, independent of disease severity (9). Furthermore, M. Lavania et al. (34) reported an average SARS-CoV-2 RNA shedding duration of approximately 14–21 days, with viral loads ranging from $10^2$ to $10^8$ RNA copies per gram. Similarly, a review of 48 studies reported that SARS-CoV-2 could be detected in feces for at least 2 weeks after the decline of symptoms (35). These findings contrast with our findings of viral clearance in fecal samples within 4 weeks. The shorter shedding duration in this cohort could reflect a combination of effective immune clearance and lower viral loads in the later stages of infection.

Interestingly, one study participant exhibited different SARS-CoV-2 strains in their feces and saliva at the same collection time point, suggesting the possibility of re-infection or concurrent infection by multiple strains. This finding aligns with a case study in which a patient acquired a second distinct SARS-CoV-2 strain during a hospital stay, resulting in an infection of the nasopharyngeal tract (36).

Our results underscore the importance of considering multiple biological samples, such as feces, in the detection and characterization of SARS-CoV-2, especially since multiple strains can be present in saliva and feces samples (Fig. S1). The presence and persistence of viral RNA in feces, even after respiratory clearance in some cases, do not necessarily indicate the presence of infectious viral particles. However, it highlights a potential role in disease transmission and persistence, as suggested by findings in immunocompromised patients who had viable SARS-CoV-2 (37). This finding is particularly relevant in clinical procedures involving patient fecal material transplant (FMT), where ensuring the absence of SARS-CoV-2 or other potential pathogens is critical to preventing transmission (38). Many FMT centers routinely screen donor stool for the absence of viral RNA. The inclusion of fecal testing in diagnostic and pre-FMT screening protocols could help mitigate risks associated with fecal–oral routes of transmission. Based on this cohort, the risk of fecal–oral transmission is higher within the 4 weeks following diagnosis, and then the risk reduces.

## Strengths and weaknesses of the study

A major strength of this study is the comprehensive sampling approach, with a large population-based cohort of 366 fecal and 365 saliva samples collected at four distinct time points from both community and hospitalized participants. Additionally, the high follow-up rate, with 80% of participants providing complete sample sets over the study period, enhances the reliability and representativeness of our findings. Another strength lies in the longitudinal design, which allows for a nuanced understanding of viral shedding dynamics over time in both feces and saliva. By monitoring participants at multiple stages of infection, our study captures insights into the persistence and clearance of SARS-CoV-2, particularly with regards to feces, a less commonly studied aspect of COVID-19 transmission potential. The inclusion of both community and hospital settings broadens the generalizability of our findings, offering insights applicable to various healthcare and community settings.

Nevertheless, there are limitations to this study. First, random sampling was not possible since there was no available sampling frame of diagnosed participants. Second, our data were collected relatively late following a positive diagnosis, with a median time of 13 days between a positive test and the first sample, consequently limiting our ability

to provide precise estimates of prevalence close to the index test date. Third, the number of unvaccinated participants in our study was small, limiting our ability to draw strong conclusions about the impact of vaccination status on viral persistence. Finally, although RT-qPCR can detect as low as 50 viral particles per 100 mg of feces (17), the detection is highly variable as inherent components of the stool, such as mucus and fiber, are known to inhibit the detection of the virus.

## Conclusions

Our findings indicate that SARS-CoV-2 RNA is detectable in fecal and saliva samples shortly after diagnosis but declines rapidly over the course of the infection, with detection more common in those with more concurrent symptoms. This pattern highlights the variability of viral shedding dynamics in different bodily fluids and underscores the potential role of symptom duration in viral persistence.

This study adds unique insights by analyzing SARS-CoV-2 shedding within a UK-based cohort from Norfolk, addressing a gap in the literature, as most existing research has focused on Asian and North American cohorts. By providing data specific to a UK demographic, our research contributes to a more geographically diverse understanding of SARS-CoV-2 shedding, supporting comparisons across populations that may experience different viral shedding profiles due to factors such as genetic background, immune response, and healthcare practices.

Our results, while broadly consistent with previous findings, demonstrate some differences in the duration of viral shedding compared to studies from other regions. This underscores the importance of conducting similar studies across diverse populations to understand the global picture of SARS-CoV-2 shedding and persistence. These findings contribute to our understanding of SARS-CoV-2 dynamics in various bodily fluids and have important implications for public health measures, patient management strategies, and the potential utility of fecal and saliva testing in monitoring the course of infection.

## ACKNOWLEDGMENTS

The authors want to thank Melinda Mayer, Sharlize Pedroza-Matute, Bushra Schuite-maker, Marnie Barham, Stephanie Ong, Jacob Scadden, and Jade Davies for helping with assembling the collection kits and capturing metadata from the anonymized question-naires. We would like to thank Dr. Andreas Brodbeck for assisting with recruiting study participants at JPUH. We thank Carmen Walker, Clare Ferns, Sarah Wilford, and Judith Gowlett for helping with consenting the participants. We thank Sumeet Tiwari and Than Le-Viet for uploading viral genomes to the public repository. We thank the participants for donating samples.

This work was supported by funding from the QIB, the Biotechnology and Biological Sciences Research Council (BBSRC) Impact Accelerator Account (BB/S506679/1), Institute Strategic Programme Gut Microbes and Health (BB/R012490/1) and its constituent projects BBS/E/F/000PR10355 and BBS/E/F/000PR10356, Institute Strategic Programme Microbes in the Food Chain BB/R012504/1 and its constituent projects BBS/E/F/000PR10348, BBS/E/F/000PR10349, BBS/E/F/000PR10351, and BBS/E/F/000PR10352. E.G.G. was funded by a Beatriz Galindo scholarship from the Spanish Ministry of Universities (BG22/00060). L.S. was supported by a BBSRC Discovery Fellowship (BB/Z514445/10).

L.K., A.N., L.S., E-G-G., C.H. (QIB), S.R., A.H., and G.S. designed the study. L.K. and C.H. (JPUH) consented the human participants and collected, or organized the collection, of the samples. J.S., A.A., D.A.Y., D.B., and L.S. inactivated samples, extracted nucleic acids, and did qPCR. L.O.M. did the phylogenetic analysis. L.S. coordinated the analysis of the samples. L.S., G.S., and E.G.-G. wrote the manuscript. All authors approved the final version of this manuscript.

## AUTHOR AFFILIATIONS

[1]Quadram Institute Bioscience, Norwich Research Park, Norwich, United Kingdom

[2]Department of Agronomic Engineering-ETSIA, Universidad Politécnica de Cartagena, Paseo Alfonso XIII, Cartagena, Region of Murcia, Spain

[3]James Paget University Hospitals NHS Foundation Trust, Great Yarmouth, England, United Kingdom

[4]Norfolk and Norwich University Hospitals NHS Foundation Trust, Norwich, United Kingdom

## AUTHOR ORCIDs

Lee Kellingray http://orcid.org/0000-0002-6135-6321
George M. Savva http://orcid.org/0000-0001-9190-124X
Enriqueta Garcia-Gutierrez http://orcid.org/0000-0001-5683-7924
Daniel Alejandro Yara http://orcid.org/0000-0003-0482-7059
Leonardo de Oliveira Martins http://orcid.org/0000-0001-5247-1320
Chloe Hutchins http://orcid.org/0000-0001-7797-9833
Ngozi Elumogo http://orcid.org/0009-0003-7354-3532
Arjan Narbad http://orcid.org/0000-0003-2968-7558
Lizbeth Sayavedra http://orcid.org/0000-0001-5814-9471

## FUNDING

| Funder | Grant(s) | Author(s) |
| --- | --- | --- |
| Biotechnology and Biological Sciences Research Council | Impact Accelerator Account (BB/S506679/1) | Lee Kellingray |
| | | George M. Savva |
| | | Enriqueta Garcia-Gutierrez |
| | | Jemma Snell |
| | | Stefano Romano |
| | | Daniel Alejandro Yara |
| | | Annalisa Altera |
| | | Leonardo de Oliveira Martins |
| | | Chloe Hutchins |
| | | David Baker |
| | | Antonietta Hayhoe |
| | | Arjan Narbad |
| | | Lizbeth Sayavedra |
| Biotechnology and Biological Sciences Research Council | BB/R012490/1: BBS/E/F/000PR10355 and BBS/E/F/000PR10356 | Lee Kellingray |
| | | George M. Savva |
| | | Enriqueta Garcia-Gutierrez |
| | | Jemma Snell |
| | | Stefano Romano |
| | | Daniel Alejandro Yara |
| | | Annalisa Altera |
| | | Leonardo de Oliveira Martins |
| | | Chloe Hutchins |
| | | David Baker |
| | | Antonietta Hayhoe |
| | | Arjan Narbad |
| | | Lizbeth Sayavedra |

| Funder | Grant(s) | Author(s) |
|---|---|---|
| Biotechnology and Biological Sciences Research Council | BB/R012504/1 | Lee Kellingray |
| | | George M. Savva |
| | | Enriqueta Garcia-Gutierrez |
| | | Jemma Snell |
| | | Stefano Romano |
| | | Daniel Alejandro Yara |
| | | Annalisa Altera |
| | | Leonardo de Oliveira Martins |
| | | Chloe Hutchins |
| | | David Baker |
| | | Antonietta Hayhoe |
| | | Arjan Narbad |
| | | Lizbeth Sayavedra |
| Biotechnology and Biological Sciences Research Council | BBS/E/F/000PR10348 BBS/E/F/000PR10349 BBS/E/F/000PR10351 and BBS/E/F/000PR10352 | Lee Kellingray |
| | | George M. Savva |
| | | Enriqueta Garcia-Gutierrez |
| | | Jemma Snell |
| | | Stefano Romano |
| | | Daniel Alejandro Yara |
| | | Annalisa Altera |
| | | Leonardo de Oliveira Martins |
| | | Chloe Hutchins |
| | | David Baker |
| | | Antonietta Hayhoe |
| | | Ngozi Elumogo |
| | | Lizbeth Sayavedra |
| Spanish Ministry of Universities | Beatriz Galindo | Enriqueta Garcia-Gutierrez |
| Biotechnology and Biological Sciences Research Council | BB/Z514445/10 | Lizbeth Sayavedra |

## AUTHOR CONTRIBUTIONS

Lee Kellingray, Conceptualization | George M. Savva, Conceptualization, Data curation, Writing – original draft, Writing – review and editing | Enriqueta Garcia-Gutierrez, Conceptualization, Data curation, Writing – original draft, Writing – review and editing | Jemma Snell, Methodology | Stefano Romano, Conceptualization | Daniel Alejandro Yara, Methodology | Annalisa Altera, Methodology | Leonardo de Oliveira Martins, Visualization | Chloe Hutchins, Conceptualization | David Baker, Methodology | Antonietta Hayhoe, Resources | Christian Hacon, Resources | Ngozi Elumogo, Resources | Arjan Narbad, Conceptualization | Lizbeth Sayavedra, Conceptualization, Data curation, Investigation, Project administration, Supervision, Visualization, Writing – original draft, Writing – review and editing

## DATA AVAILABILITY

Viral genomes were submitted under the project PRJNA1141947.

## ETHICS APPROVAL

Human tissue samples were accessed in accordance with the research protocol approved by the Research Ethics Committees Northern Ireland (ORECNI) REC REF 20-NI-0076 and registered at clinicaltrials.gov NCT04546776 under IRAS ID 284252.

## ADDITIONAL FILES

The following material is available online.

### Supplemental Material

**Fig. S1 (Spectrum03195-24-s0001.docx).** Distribution of positive samples and variants.

### Open Peer Review

**PEER REVIEW HISTORY (review-history.pdf).** An accounting of the reviewer comments and feedback.

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
