## [Reviewer comments · Microbiology Spectrum]

Microbiology Spectrum

Temporal Dynamics of SARS-CoV-2 Shedding in Faeces and Saliva: A Longitudinal Study in Norfolk, United Kingdom during the 2021-2022 COVID-19 Waves

Lee Kellingray, George Savva, Enriqueta García Gutiérrez, Jemma Snell, Stefano Romano, Daniel Yara, Annalisa Altera, Leonardo de Oliveira Martins, Chloe Hutchins, Dave Baker, Antonietta Hayhoe, Christian Hacon, Ngozi Elumogo, Arjan Narbad, and Lizbeth Sayavedra

Corresponding Author(s): Lizbeth Sayavedra, Quadram Institute Bioscience

Review Timeline:

Submission Date:	December 6, 2024
Editorial Decision:	January 21, 2025
Revision Received:	February 18, 2025
Accepted:	February 24, 2025

Editor: Eleanor Powell

Reviewer(s): Disclosure of reviewer identity is with reference to reviewer comments included in decision letter(s). The following individuals involved in review of your submission have agreed to reveal their identity: Pablo Tsukayama (Reviewer #1)

Transaction Report:

DOI: <https://doi.org/10.1128/spectrum.03195-24>

Re: Spectrum03195-24 (Temporal Dynamics of SARS-CoV-2 Shedding in Faeces and Saliva: A Longitudinal Study in Norfolk, United Kingdom during the 2021-2022 COVID-19 Waves)

Dear Dr. Lizbeth Sayavedra:

Thank you for the privilege of reviewing your work. After receiving feedback from two reviewers, modifications are required before potential publication. Below you will find instructions from the Spectrum editorial office and the reviewer comments.

Revision Guidelines

Sincerely,
Eleanor Powell
Editor
Microbiology Spectrum

Reviewer #1 (Comments for the Author):

This manuscript investigates the temporal dynamics of SARS-CoV-2 shedding in faeces and saliva, using longitudinal samples collected from 98 participants in Norfolk, United Kingdom, during the 2021-2022 COVID-19 waves. Using RT-qPCR and whole-genome sequencing, the study finds that saliva samples are more likely to test positive for SARS-CoV-2 and have a longer duration than fecal samples. Viral shedding correlates positively with symptom severity, and vaccinated individuals show

significantly lower detection rates. In some cases, the authors identified distinct viral strains, highlighting tissue-specific viral dynamics.

The authors' conclusions are well-supported by their data. The statistical analyses and sequencing results substantiate the observations of differential shedding dynamics in feces and saliva. The association between symptom severity and viral detection, as well as the impact of vaccination, is appropriately analyzed using logistic regression models. Moreover, the authors' methodology, which includes RT-qPCR detection, phylogenetic analysis, and multilevel logistic regression, provides a solid basis for the conclusions drawn.

The manuscript is written in clear, standard English and is easy to understand to this non-native reviewer. While minor grammatical edits could improve readability further, these are not critical and do not detract from the clarity of the manuscript.

The study includes SARS-CoV-2 WGS data, which have been deposited in a public repository under project PRJNA1141947.

This study has several strengths, including a longitudinal design with high follow-up rates (81%), which enables precise insights into viral shedding dynamics. The integration of molecular methods such as RT-qPCR and sequencing strengthens the reliability of the findings. However, the delayed recruitment (~13 days post-diagnosis) limits conclusions about viral shedding in the very early stages of infection. Additionally, the small number of unvaccinated participants reduces the statistical power to fully assess the impact of vaccination.

To improve the manuscript, the authors could include a brief discussion on how their findings align or diverge from global studies, particularly in relation to population-specific factors such as genetic background or healthcare practices. Adding a table summarizing key differences in shedding dynamics between saliva and faeces would also enhance clarity for readers.

Overall, this manuscript is technically sound and methodologically rigorous, meeting the standards of Microbiology Spectrum. It provides valuable insights into SC-2 shedding dynamics in saliva and faeces, with implications for public health surveillance and clinical practices.

Reviewer #2 (Comments for the Author):

Thank you for the opportunity to review the manuscript entitled " Temporal Dynamics of SARS-CoV-2 Shedding in Faeces and Saliva: A Longitudinal Study in Norfolk, United Kingdom during the 2021-2022 COVID-19 Waves". In summary, Kellingray et al., investigated the longitudinal presence of SARS-CoV-2 nucleic acid and genomic sequences in faecal and saliva samples from patients with covid. Interestingly, they found that saliva was positive for longer than faeces post-infection and detection in faeces was associated with number of reported (and severity of?) symptoms. This study is relevant and interesting. I have a few comments, questions and suggestions.

Line 143-152 Were negative controls used? If so, which sample types? It may be necessary to demonstrate that un-infected patients do not have detectable levels of SARS-CoV-2 in their faeces/saliva using the same assay.

Figure 1: Do we know that patients were not re-infected? There is a dip in the percent positive at 4 weeks- did those same patients then have positives at weeks 5 and/or 6

Line 279: does symptom count equal symptom severity? Since one of the conclusions was that detection in faeces was associated with severity of symptoms, but is it actually associated with number of symptoms? How was severity determined?

Table 1: For the "ever vaccinated" characteristic for JPUH column, wondering if this should say 17 since the % is 100% and the n=17? Or is this because 10 people were missing this on their questionnaires, so $17-10 = 7$? Does that mean that 10 people didn't answer the vaccination question on all their questionnaires?

Line 361-362 Maybe this was re-infection instead of truly concurrent infection?

Line 370-372 While viral nucleic acid was detected in faeces, that may not represent truly infectious viral particles.

Line 385-397 It would also be interesting to see how the faeces and saliva detection compares with nasopharyngeal detection using the same assay. Do you think these patients would be testing positive by clinical, diagnostic nasopharyngeal testing too?

We would like to thank the two reviewers for their thoughtful and positive feedback on our study. Please find our point-by-point response below.

Reviewer #1 (Comments for the Author):

This manuscript investigates the temporal dynamics of SARS-CoV-2 shedding in faeces and saliva, using longitudinal samples collected from 98 participants in Norfolk, United Kingdom, during the 2021-2022 COVID-19 waves. Using RT-qPCR and whole-genome sequencing, the study finds that saliva samples are more likely to test positive for SARS-CoV-2 and have a longer duration than fecal samples. Viral shedding correlates positively with symptom severity, and vaccinated individuals show significantly lower detection rates. In some cases, the authors identified distinct viral strains, highlighting tissue-specific viral dynamics.

The authors' conclusions are well-supported by their data. The statistical analyses and sequencing results substantiate the observations of differential shedding dynamics in feces and saliva. The association between symptom severity and viral detection, as well as the impact of vaccination, is appropriately analyzed using logistic regression models. Moreover, the authors' methodology, which includes RT-qPCR detection, phylogenetic analysis, and multilevel logistic regression, provides a solid basis for the conclusions drawn.

The manuscript is written in clear, standard English and is easy to understand to this non-native reviewer. While minor grammatical edits could improve readability further, these are not critical and do not detract from the clarity of the manuscript.

Thank you for your feedback. We have rephrased a few sentences for improved clarity.

The study includes SARS-CoV-2 WGS data, which have been deposited in a public repository under project PRJNA1141947.

This study has several strengths, including a longitudinal design with high follow-up rates (81%), which enables precise insights into viral shedding dynamics. The integration of molecular methods such as RT-qPCR and sequencing strengthens the reliability of the findings. However, the delayed recruitment (~13 days post-diagnosis) limits conclusions about viral shedding in the very early stages of infection.

Additionally, the small number of unvaccinated participants reduces the statistical power to fully assess the impact of vaccination.

We have addressed the limitations in the discussion section of the study. Thank you for highlighting these aspects.

To improve the manuscript, the authors could include a brief discussion on how their findings align or diverge from global studies, particularly in relation to population-specific factors such as genetic background or healthcare practices.

Thanks for your suggestion. We have included the following:

“Recent genomic epidemiology research in Norfolk, UK (Hayles et al., 2024) identified extensive SARS-CoV-2 lineage diversity, with shifts in dominant variants occurring on a slightly different timeline than national and global trends. Since viral genetics influence shedding dynamics, variations in lineage predominance may have contributed to the shedding durations observed in our study. Factors such as age, disease severity, and immune response, as well as population-specific dynamics

such as vaccination coverage, healthcare interventions, and demographic characteristics (e.g. Norfolk's older population compared to the rest of the UK), including re-infection, could account for the variability in shedding duration reported across studies." (Line 368-372)

Adding a table summarizing key differences in shedding dynamics between saliva and faeces would also enhance clarity for readers.

We appreciate the suggestion. A table summarizing the key differences in shedding dynamics between saliva and faeces is already included (Table 2), which presents the number and proportion of positive samples stratified by the number of completed weeks since the index positive test.

Overall, this manuscript is technically sound and methodologically rigorous, meeting the standards of Microbiology Spectrum. It provides valuable insights into SC-2 shedding dynamics in saliva and faeces, with implications for public health surveillance and clinical practices.

Reviewer #2 (Comments for the Author):

Thank you for the opportunity to review the manuscript entitled " Temporal Dynamics of SARS-CoV-2 Shedding in Faeces and Saliva: A Longitudinal Study in Norfolk, United Kingdom during the 2021-2022 COVID-19 Waves". In summary, Kellingray et al., investigated the longitudinal presence of SARS-CoV-2 nucleic acid and genomic sequences in faecal and saliva samples from patients with covid. Interestingly, they found that saliva was positive for longer than faeces post-infection and detection in faeces was associated with number of reported (and severity of?) symptoms. This study is relevant and interesting. I have a few comments, questions and suggestions.

Line 143-152 Were negative controls used? If so, which sample types? It may be necessary to demonstrate that un-infected patients do not have detectable levels of SARS-CoV-2 in their faeces/saliva using the same assay.

Thank you for your feedback. We optimised the methodology of this work in a previous study¹, where saliva and faecal samples collected before the pandemic were spiked with increasing amounts of SARS-CoV-2, starting with no spike. The samples without the spiked virus consistently tested negative.

Figure 1: Do we know that patients were not re-infected? There is a dip in the percent positive at 4 weeks- did those same patients then have positives at weeks 5 and/or 6

A small number of the participants had a negative sample and then a positive. We have included the possibility of re-infection, as well as possible changes in the viral particles across time as follows (Line 304-306):

"This may be due to re-infection during the sampling period, or could indicate that the level of viral particles present in the samples was lower than the limit of detection of our assay."

Line 279: does symptom count equal symptom severity? Since one of the conclusions was that detection in faeces was associated with severity of symptoms, but is it actually associated with number of symptoms? How was severity determined?

Thank you for pointing this out. We have revised the manuscript to refer to "number of symptoms" instead of "severity" throughout. However, we would expect the number of symptoms to generally correlate with severity.

Table 1: For the "ever vaccinated" characteristic for JPUH column, wondering if this should say 17 since the % is 100% and the n=17? Or is this because 10 people were missing this on their questionnaires, so $17-10 = 7$? Does that mean that 10 people didn't answer the vaccination question on all their questionnaires?

Apologies, we agree that this was not clear but the percentage is the proportion of those for whom we have the data. The table description has been amended to make this clearer -

“**Table 1.** Characteristics of participants included in the study. The percentages in this table were calculated based on the number of individuals who returned their questionnaires.”

Line 361-362 Maybe this was re-infection instead of truly concurrent infection?

Excellent point. We have rephrased the text as follows (Line 386-388):

“Interestingly, one study participant exhibited different SARS-CoV-2 strains in their faeces and saliva at the same collection time point, suggesting the possibility of re-infection or concurrent infection by multiple strains.”

Line 370-372 While viral nucleic acid was detected in faeces, that may not represent truly infectious viral particles.

We have acknowledged that the presence of the virus may not represent truly infectious particles as follows (Line 393-397):

The presence and persistence of viral RNA in faeces, even after respiratory clearance in some cases, does not necessarily indicate the presence of infectious viral particles. However, it highlights a potential role in disease transmission and persistence, as suggested by findings in immunocompromised patients who had viable SARS-CoV-2².

Line 385-397 It would also be interesting to see how the faeces and saliva detection compares with nasopharyngeal detection using the same assay. Do you think these patients would be testing positive by clinical, diagnostic nasopharyngeal testing too?

We appreciate the reviewer's interest in understanding how faecal and saliva detection of SARS-CoV-2 compares with nasopharyngeal detection using the same assay. Based on existing literature (e.g.,³), nasal swabs and saliva samples are increasingly considered viable alternatives to nasopharyngeal swabs for detecting SARS-CoV-2, particularly due to their ease of collection and patient comfort.

In a study comparing nasal swabs, saliva samples, and nasopharyngeal swabs, it was found that while nasopharyngeal swabs exhibited the lowest cycle threshold (Ct) values—indicating the highest viral concentrations—saliva samples were still able to yield positive results for SARS-CoV-2. This study suggested that saliva and nasal samples may serve as alternative diagnostic specimens in situations where nasopharyngeal swabs are not feasible or preferred.

Therefore, it is likely that patients who tested positive through saliva detection in our study would also have tested positive using clinical nasopharyngeal diagnostic testing, considering that saliva samples have shown comparable sensitivity under rigorous collection conditions.

However, as we did not collect nasopharyngeal swabs in our study, we are unable to directly confirm this for our cohort.

References

- 1 Li, T. *et al.* An optimised protocol for detection of SARS-CoV-2 in stool. *BMC Microbiology* **21**, 1-8 (2021).
- 2 Dergham, J., Delerce, J., Bedotto, M., La Scola, B. & Moal, V. Isolation of viable SARS-CoV-2 virus from feces of an immunocompromised patient suggesting a possible fecal mode of transmission. *Journal of Clinical Medicine* **10** (2021). <https://doi.org:10.3390/jcm10122696>
- 3 Jung, E. J. *et al.* Comparison of nasal swabs, nasopharyngeal swabs, and saliva samples for the detection of SARS-CoV-2 and other respiratory virus infections. *Ann Lab Med* **43**, 434-442 (2023). <https://doi.org:10.3343/alm.2023.43.5.434>

Re: Spectrum03195-24R1 (Temporal Dynamics of SARS-CoV-2 Shedding in Faeces and Saliva: A Longitudinal Study in Norfolk, United Kingdom during the 2021-2022 COVID-19 Waves)

Dear Dr. Lizbeth Sayavedra:

It is my pleasure to inform you that your manuscript has been accepted, and I am forwarding it to the ASM production staff for publication. Your paper will first be checked to make sure all elements meet the technical requirements. ASM staff will contact you if anything needs to be revised before copyediting and production can begin. Otherwise, you will be notified when your proofs are ready to be viewed.

Sincerely,
Eleanor Powell
Editor
Microbiology Spectrum